# Heteroatom-Doped Porous Carbon-Based Nanostructures for Electrochemical CO_2_ Reduction

**DOI:** 10.3390/nano12142379

**Published:** 2022-07-12

**Authors:** Qingqing Lu, Kamel Eid, Wenpeng Li

**Affiliations:** 1Engineering & Technology Center of Electrochemistry, School of Chemistry and Chemical Engineering, Qilu University of Technology (Shandong Academy of Sciences), Jinan 250353, China; qqlu@qlu.edu.cn (Q.L.); liwenpeng@qlu.edu.cn (W.L.); 2Gas Processing Center (GPC), College of Engineering, Qatar University, Doha 2713, Qatar; 3Shandong Key Laboratory of Biochemical Analysis, College of Chemistry and Molecular Engineering, Qingdao University of Science and Technology, Qingdao 266042, China

**Keywords:** doped carbon, heteroatom, porous carbon CO_2_ reduction, CO_2_ conversion, metal-free electrocatalysts, electrochemical CO_2_ reduction

## Abstract

The continual rise of the CO_2_ concentration in the Earth’s atmosphere is the foremost reason for environmental concerns such as global warming, ocean acidification, rising sea levels, and the extinction of various species. The electrochemical CO_2_ reduction (CO_2_RR) is a promising green and efficient approach for converting CO_2_ to high-value-added products such as alcohols, acids, and chemicals. Developing efficient and low-cost electrocatalysts is the main barrier to scaling up CO_2_RR for large-scale applications. Heteroatom-doped porous carbon-based (HA-PCs) catalysts are deemed as green, efficient, low-cost, and durable electrocatalysts for the CO_2_RR due to their great physiochemical and catalytic merits (i.e., great surface area, electrical conductivity, rich electrical density, active sites, inferior H_2_ evolution activity, tailorable structures, and chemical–physical–thermal stability). They are also easily synthesized in a high yield from inexpensive and earth-abundant resources that meet sustainability and large-scale requirements. This review emphasizes the rational synthesis of HA-PCs for the CO_2_RR rooting from the engineering methods of HA-PCs to the effect of mono, binary, and ternary dopants (i.e., N, S, F, or B) on the CO_2_RR activity and durability. The effect of CO_2_ on the environment and human health, in addition to the recent advances in CO_2_RR fundamental pathways and mechanisms, are also discussed. Finally, the evolving challenges and future perspectives on the development of heteroatom-doped porous carbon-based nanocatalysts for the CO_2_RR are underlined.

## 1. Introduction

The incessant utilization of fossil fuels as energy sources as well as industrialization result in greenhouse gas emissions (GHGs) and other hazardous gases that are the main reason for the environmental catastrophes that threaten life on planet Earth, such as climate change and global warming [1,2,3,4]. Among GHGs, CO_2_ alone contributes ~20%; the CO_2_ emission level increased from ~390 ppm in 2012 to 420 ppm in 2021, but nearly 1% of this amount is being removed annually [1]. Finding green energy resources (i.e., fuel cells [5,6,7,8,9,10], solar cells [11,12], water electrolysis [13,14,15,16], and batteries [17,18]), CO_2_ capture [19,20], and CO_2_ conversion [1] are the main approaches to reducing CO_2_ levels in the Earth’s atmosphere. CO_2_ can be easily converted to high-value-added chemicals and fuels (i.e., alcohols, acids, CO, and methane) using reforming (i.e., steam and dry), photocatalytic, and biological CO_2_RR [21,22,23,24,25]. The electrochemical CO_2_RR is promising as an efficient and green approach, owing to its ambient operating conditions (i.e., room temperature, atmospheric pressure, controllable production of various products under adjustable potential) [21,22,23]. This is driven by various metals/oxides (i.e., Cu, Ru, Ir, Rh, TiO_2_, SnO_2_, and CeO_2_), metal chalcogenides (i.e., ZnTe, SnS_2_, and CdS), carbon nitrides, metal–organic frameworks, and perovskites [26,27,28,29,30,31,32].

Unlike these catalysts, carbon materials are deemed as greener, efficient, low-cost catalysts feasible for practical CO_2_RR [33,34,35,36,37]. Heteroatom (i.e., N, P, O, B, halogens)-doped porous carbon nanostructures (HA-PCs) possess many advantages compared with other carbon-based catalysts. This is owing to their intrinsic physicochemical merits (i.e., metal-free nature, chemical durability, thermal stability), unique catalytic merits (i.e., outstanding surface area to volume ratio, high electrical conductivity, rich electrical density, accessible active sites, low H_2_ evolution activity, and tailorable structures), and distinctive properties of porous structures (i.e., maximized atom utilization, quick electron mobility, ease of gas adsorption, and diffusion) [33,38,39,40,41,42,43,44,45]. HA-PCs could be easily produced in a high yield (i.e., kilogram scale) from green, earth-abundant, and cheap resources (i.e., carbohydrate, cellulose, and lignin) that can meet the sustainability and large-scale application requirements [44,46,47,48]. Thereby, the engineering of porous carbon materials for CO_2_RR has attracted significant attention in the last decade, resulting in nearly 750 articles; meanwhile, 53 are exclusively dedicated to HA-PCs for CO_2_RR (Figure 1a). Accordingly, it is imperative to provide a timely update on this theme of CO_2_RR executed on HA-PCs. Notably, various reviews have been published for the CO_2_RR using carbon-based catalysts; however, they were not all related to HA-PCs (Table 1) [34,49,50,51,52,53].

In pursuit of this aim, this review highlights the engineering approaches of HA-PCs for the electrochemical CO_2_RR in addition to the effect of mono, binary, and ternary heteroatoms (i.e., N, S, F, or B) on the CO_2_RR activity, durability, and selectivity from both experimental and theoretical views (Figure 1b). This is in addition to underlining the CO_2_RR pathway and mechanisms along with the current challenges and perspectives on developing heteroatom-doped porous carbon nanocatalysts for the CO_2_RR.

## 2. Effect of CO_2_ on the Environment and Human Health

CO_2_ emissions come from industrial activities (32%), building operations (28%), transportation (23%), building materials and construction (11%), and other sources (6%) and are one of the main reasons for global warming and climate change, which affect humans and the environment significantly [1,55]. CO_2_ absorbs a lower heat than other greenhouse gases, but it remains in the atmosphere longer and acts as a blanket in the air, trapping heat in the atmosphere and warming up the Earth’s temperature and increasing the temperature of the ocean (0.07 °C/decade) [1,55]. Increases in Earth’s temperature lead to several environmental changes (i.e., shrinking H_2_O supplies, geographic weather patterns, food supplies, acid rains, and sea level). Moreover, CO_2_ reacts with the H_2_O of the ocean to produce carbonic acid, which decreases the ocean’s pH, reducing the ability of marine life to extract calcium from the water to build their shells [55]. Humans are exposed daily to CO_2_ indoors and outdoors, including inside homes, workplaces, and streets; CO_2_ is a product of human metabolism and is respired into the ambient air. The average CO_2_ indoors ranges from 600 to 1000 ppm but can reach 2000 ppm with increased room occupancies and poor ventilation rates [56]. Thus, the effect of CO_2_ on human health should usually be emphasized and studied, as was extensively discussed in another review [56]. Excessive exposure to CO_2_ can lead to increased heart rate, blood pressure, difficulty breathing, brain damage, coma, and even death, depending on CO_2_ concentration and exposure time [57]; Figure 1c depicts the effect of CO_2_ on human health as a factor of concentration and exposure time. The concentration of CO_2_ indoors or outdoors should not exceed 1000 ppm for more than 2.5 h to avoid any hazardous effects. Meanwhile, the dangerous effects of exposure to CO_2_ comprise inflammation at 2000–4000 ppm for 2 h, cognitive effects at 1000–2700 ppm for 1–6 h, bone demineralization/kidney calcification at 2000–3000 ppm for 60–90 days, behavioral changes/physiological stress at 700–3000 ppm for 13–15 days, oxidative stress/endothelial dysfunction at 3000–5000 ppm for 13 days to 6 months, and brain damage or death at 40,000 ppm [56,57]. Interestingly, it has been revealed that CO_2_ can be stored inside the human body, as indicated by the significant increase in serum bicarbonate in the general US population [58,59]. Few studies have reported the direct effect of CO_2_ on human health; therefore, more experimental and theoretical studies are needed [56].

## 3. Fundamental Parameters for CO_2_RR Performance

The fundamental parameters for estimating the CO_2_RR performance are the Faradaic efficiency (FE), overpotential, current density, durability, and energy efficiency [1,53].

### 3.1. Faradaic Efficiency (FE)

The FE is calculated from the following equation:FE = nmF/Q,
where n, m, F, and Q are the number of electrons, total mole amounts of the product, Faraday constant (96,485 C/mol), and amounts of cumulative charge during CO_2_ reduction. The FE determines the CO_2_RR selectivity for various gas/liquid products (i.e., CH_4_, CO, HCOOH, and CH_3_OH), so a higher FE for specific products is highly required to reduce the cost of isolation products. A catalyst with an inferior HER ability is desired to yield a great FE and high selectivity, which could be enhanced via the integration of heteroatom dopants into the carbon skeleton structure and modifying the hydrophobicity of electrodes. Organic and hybrid electrolytes can also improve the FE due to their great CO_2_ adsorption ability and higher solubility of CO_2_ during CO_2_RR; however, the effect of electrolytes on CO_2_RR activity and selectivity is not yet resolved and remains ambiguous [34,49,50,51,52,53].

### 3.2. Overpotential

The CO_2_RR half-reaction under applied potential, which is a nonspontaneous process driven by more negative potentials than standard potentials in actual electrocatalytic conditions, results in various products. The difference between the standard thermodynamic potential of a specific half-reaction and the applied potential in the half-cell reaction is the overpotential. This is often used to activate inert CO_2_ molecules to form a bent *CO_2_^•−^ anion radical and allow electron transport during the CO_2_RR process. The lower overpotential is desired to mimic the practical process and reduce the CO_2_RR production cost, which could be achieved by modulating carbon-based catalysts’ morphology and composition. The CO_2_RR can occur through a single and multiple proton-coupled electron transfer process, so carbon-based materials with tunable structures and properties are thermodynamically feasible for single electron and multiple electron transfer processes [34,49,50,51,52,53].

### 3.3. Partial Current Density

The partial current density (j_p_) represents the current density needed for the production of specific products, and it is calculated using the following equation:j_p_ = j × FE
where j is the total current density, so the effective catalyst should deliver a high j_p_ under a low applied potential. The j_p_ depends on the inherited catalytic properties of HA-PCs (i.e., electrical conductivity, interaction with the electrode, and CO_2_ adsorption) in addition to the electrolyte and cell design [34,49,50,51,52,53]. The flow cell system allows prompt CO_2_ feeding into the cathode with the assistance of a gas diffusion electrode, resulting in an outstanding j_p_. Notably, to date, the obtained total current density is lower than 1 A, which is still far beyond commercial requirements [34,49,50,51,52,53].

### 3.4. Durability

The durability of FE and current density are some of the essential factors hinging on CO_2_RR with maintained selectivity and activity under prolonged potentiostatic polarization. Notably, durability tests are carried out for a few hours, which is insufficient for the large-scale CO_2_RR applications; however, currently, most of the catalysts are exposed to significant attenuation in the current density, FE, and selectivity, owing to blocking and deactivation of the active catalytic sites and structural degradation. In light of the commercialization scale, the stability tests should be conducted for several weeks or months (≥1000 h) along with carrying out various in situ and ex situ characterizations to understand the degradation mechanism and solve this critical issue. Carbon-based catalysts, with their impressive chemical–physical stability, are promising to solve the stability issues in CO_2_RR, but they are not investigated enough. The modification of the electrochemical cell to allow continued CO_2_ flow, refreshing the electrolyte solution, preventing gas accumulation over the cathode, and gathering liquid products, can enhance the long-term stability of porous carbon-based catalysts [34,49,50,51,52,53].

### 3.5. Energy Efficiency (E_eff_)

The *E*_eff_ is a crucial factor in estimating the economic efficiency of CO_2_RR as it is the percentage of the energy stored in the chemical. The *E*_eff_ is calculated based on the equilibrium potential (*E*_eq_) using the following equation:

*E*_eff_ = [*E*_eq_/*E*_eq_ + *η*] × EF

Thus, a great *E*_eff_ together with outstanding catalytic activity drives the CO_2_RR half-reaction at a low *η*, negligible ohmic potential drop, and high FE. The high *E*_eff_ is optimized via adjusting the electrolyte, membrane, and cell structure in addition to carbon-based electrode conductivity. Notably, the relationship between the structure/composition of HA-PC catalysts and *E*_eff_ is not yet studied [34,49,50,51,52,53].

### 3.6. Turnover Frequency (TOF)

TOF is the amount for generating specific products over single active sites per unit time during CO_2_RR process, so it is an imperative factor toward the intrinsic activity of HA-PCs. The TOF is calculated from the following equation:TOF = *N*_p_/*N*_c_
where *N*_p_ and *N*_c_ are the product’s mole number and the mole number of the catalyst’s active site, respectively. A high TOF value indicates the presence of multiple active sites. Porous carbon-based doped materials usually possess a high TOF for gas products, especially CO, but the effect of the shape and composition of porous carbon-based catalysts with heteroatoms has not yet been emphasized [34,49,50,51,52,53].

## 4. Engineering Methods of Heteroatom-Doped Porous Carbon

There are limited approaches for the fabrication of template-based, activation, element-doping, and direct annealing of biomass-based resources, which vary in their productivity for controlling porosity (i.e., pore-volume, pore ordering, and pore size) and surface area [34,49,50,51,52,53].

### 4.1. Template-Based Method

The template-based method is the most effective approach for the rational synthesis of HA-PCs with well-defined morphology, surface area, and ordered porosity driven by the template shape, structure, and properties. There are various hard templates (i.e., MgO, AlO, CaCO_3_, ZnO, and SiO_2_), soft templates (i.e., surfactants, polymers, and ionic liquids), and self-templates (i.e., biomass and metal–organic frameworks (MOFs)) or their hybrids [60,61,62,63]. The carbon precursors should be initially infiltrated into the template and annealed at an elevated temperature, followed by chemical etching of the template in acid or alkaline solutions. The source for heteroatoms can be mixed initially with the carbon source to produce heteroatom-doped porous carbon nanostructures.

#### 4.1.1. Hard Templates

With its unique physiochemical properties, the ZnO template can generate porous carbon with 1D (i.e., nanorods and nanotubes) and 3D nanostructures via the incorporation of activators (i.e., KOH). ZnO is easily prepared commercially at a low cost via Zn’s electrochemical anodic oxidation method under ambient conditions. Moreover, carbon precursors react with ZnO at a high temperature to produce CO_2_ gas that acts as an activator to enhance the porosity and surface area. Notably, Zn tends to evaporate via an annealing process at 900 °C, resulting in eliminating the need for acids or alkali solution to remove the ZnO template. Porous core/shell carbon microrods with a high surface area of 660 m^2^/g were prepared using a ZnO microrod template and glucose as a carbon source (Figure 2a–c) [64].

ZnO microrods on Ni-foil were initially designed by chemical bath deposition, then coated with Ni metals by electro-deposition, and then coated with glucose via autoclave at 180 °C for 3.5 h, followed by annealing at 500 °C under Ar (Figure 2a–c). At 500 °C, glucose is carbonized into carbon nanospheres coated with porous microrods formed from ZnO’s evaporation. Three-dimensional flower-like hierarchical porous carbon nanostructures with a surface area of 761.5 m^2^/g, pore volume of 0.49 cm^3^/g, and microporosity of 49% were prepared via a ZnO template in the presence of HO as an activator [67]. With its unique crystal structure, MgO acts as a template for producing porous 2D sheets, 3D clusters, and 2D/3D nanostructures [68]. Mesoporous 3D carbon nanosheets with a BET surface area (883 m^2^/g) and pore size (6–8 nm) were synthesized using a MgO template and coal tar pitch as the carbon source, followed by washing with 10% HCl to remove MgO [69]. Porous carbon nanocages (CNC670) with a surface area of 2053 m^2^/g and pore size of 3–7 nm were obtained using MgO as the template and benzene as a carbon source via annealing at 670 °C and then removal of MgO by HCl [70]. Notably, annealing at a high temperature of 700, 800, and 900 °C led to decreasing the surface area to 1854, 1633, and 312 m^2^/g, respectively. Other Mg-based materials could also be used as a template for the production of HA-PCs, because they can produce in situ MgO templates. Hierarchical porous carbon nanosheets with a surface area of 2300 m^2^/g were formed using starch as a reductant and carbon source while Mg(NO_3_)_2_ was an oxidant and in situ provided a MgO template [71]. Similarly, layered porous carbon nanosheets with a surface area of 1312 m^2^/g were formed via the combustion of pectin with Mg(NO_3_)_2_ [72]. Three-dimensional carbon nanocage networks with a surface area of 1470–1927 m^2^ g^−1^ and pore size of 15–24 nm were formed via annealing of the pitch with Mg_5_(CO_3_)_4_ and KOH treatment [73]. Porous carbon sheets with a surface area of 3145 m^2^/g and abundant micropores were formed via annealing of coal tar pitch with Mg(OH)_2_ and in situ KOH activation [74]. Carbon nanocages with a surface area of 3368 m^2^/g and pore volume of 1.7 cm^3^/g were formed using Mg metal and CO_2_ as precursors [75].

There are various types of SiO_2_-based templates (FSM-16, MCM-48, KIT-6, and SBA-15) that can drive porous carbon nanostructure formation with a well-defined shape and porosity. Co/N-doped mesoporous 3D carbon nanostructures with a surface area of 568 m^2^/g and pore size of ~12 nm were formed using colloidal SiO_2_ as a template and vitamin B12 as a template (VB12/Silica colloid) (Figure 2d–i) [65]. Under the same condition, using SBA-15 produced porous nanorods (VB12/SBA-15) with a surface area of 387 m^2^/g and pore diameter of 3.5 nm, while 2D carbon nanosheets with a surface area of 239 m^2^/g and pore size of 4.5 nm were obtained using montmorillonite silica as a template (VB12/MMT) (Figure 2d–i) [65]. N-doped porous carbon nanosheets with a surface area of 1676 m^2^/g and large pore volume of 2.13 cm^3^/g were formed via SiO_2_ spheres as a template and polyaniline as the C/N source, followed by KOH activation at 850 °C [76]. Without KOH activation, the same group synthesized 2D mesoporous carbon layered nanosheets with a surface area of 582.7 m^2^/g using mesoporous SiO_2_ nanoplates as a template and coal tar pitch as the carbon source, which implies the significant role of activation on the enhancement of the pore volume and surface area [77]. Si-based precursors such as tetraethylorthosilicate (TEOS) could be used to allow in situ formation of SiO_2_ via the hydrolysis process and could act as in situ templates for the production of porous carbon [78]. A mesoporous porous carbon microsphere with a surface area of 659–872 m^2^/g and mesopore diameter of 3.2–14 nm was fabricated using TEOS within resorcinol–formaldehyde polymer microspheres and NH_4_OH as a catalyst, followed by annealing [79].

#### 4.1.2. Ca-Based Templates

CaCO_3_ can act as a template and activator due to its ability to decompose at 500 °C to release CO_2_ (activator); the CaO template forms hierarchical porous carbon with multiple pores (i.e., micro-, meso-, and macropores) after the removal of CaO. Hierarchical porous carbon nanofibers (HPCNFs-3-1) with a surface area of 679 m^2^/g and pore volume of 0.41 cm^3^/g were prepared via the electrospinning of polyacrylonitrile (PAN) (as the C/N source)/N, N’-dimethylformamide (DMF)/tetrahydrofuran (THF) with nano-CaCO_3_ (as a template) followed by carbonization at 800 °C for 2 h and then template removal by 1.5 M HCl (Figure 2j) [66]. Under annealing, nano-CaCO_3_ decomposes to release CO_2_ that produces microspores and mesopores. Then, CaO removal by HCl creates macropores and changes the ratios of PAN/DMF/THF, having an insignificant effect on the surface area and pore volume of the resultant porous carbon fibers. N-doped hierarchical porous carbon with a surface area of 1091 m^2^/g, pore volume of 0.52 cm^3^/g, and N content of 10.59% was synthesized via annealing of gelatin and graphene oxide with CaCO_3_ at 900 °C and then activation by KOH. The activation substantially affected the surface area and pore volume because, without activation, they decreased to 433 m^2^ g and 4.5 cm^3^/g, respectively [80]. Cornstalk rind (whole plant) was annealed with CaCO_3_ and K_2_C_2_O_4_ (activator) at 800 °C to hierarchical porous carbon with a surface area of 2054 m^2^/g and pore volume of 1.382 cm^3^/g [81]. The surface area (1419–2054 m^2^/g) and pore volume (0.3704–1.382 cm^3^/g) were regulated via adjustment of the activation ratio by K_2_C_2_O_4_ but without activation of the surface area (482 m^2^/g) [81]. The following equations propose the reaction between CaCO_3_ and K_2_C_2_O_4_:K_2_C_2_O_4_ → 2K_2_CO_3_ + CO
K_2_CO_3_ + CaCO_3_ → K_2_Ca(CO_3_)_2_
K_2_Ca(CO_3_)_2_ + CaCO_3_ → K_2_Ca_2_(CO_3_)_3_
CaCO_3_ → CaO + O_2_
K_2_Ca_2_(CO_3_)_3_ → K_2_Ca(CO_3_)_2_ + CaO + CO_2_
K_2_Ca(CO_3_)_2_ → K_2_CO_3_ + CaO + CO_2_
K_2_CO_3_ + 2C → 2K + 3CO

Similarly, crumpled carbon network-like nanosheets with a surface area of 1822 m^2^/g and pore volume of 4.11 cm^3^/g were formed using annealing anthracene oil with CaCO_3_ as a template coupled with KOH activation [82]. Other Ca-based sources such as Ca(NO_3_)_2_ and Ca(OH)_2_ could be used as templates and activation for directing the formation of porous carbon nanostructures, but they have rarely been reported.

#### 4.1.3. New Templates

Various new templates such as dry ice (CO_2_), MXene, and melamine can act as a template for producing HA-PCs. Dry CO_2_ produces porous carbon with a lower surface area (100 m^2^/g), so other templates or activation are needed to enhance the surface area and porosity. Mixing phenolic resin, triblock copolymer F127, and Ti_3_C_2_T*_x_* MXene followed by annealing and chlorination formed a 2D/2D porous heterostructure with a surface area of 1021 m^2^/g and a pore volume of 58% (Figure 3a) [83]. F127 and phenolic resin molecules are assembled into spherical micelles, which penetrate the interlayers of Ti_3_C_2_T*_x_* to form Ti_3_C_2_T*_x_*-micelle@resol. Then after annealing, Ti_3_C_2_T*_x_*-micelle@resol is converted into porous carbon (Ti_3_C_2_T*_x_*-OMC), and the removal of Ti in Ti_3_C_2_T*_x_*-OMC produces MXene-derived porous carbon (MDC-OMC) [83]. The chlorination of MDC-OMC at high temperature severely impacted the increment of the surface area and pore volume of MDC-OMC [83].

Hierarchical porous carbon tunable porosity (i.e., micro-, meso-, and macroporous), a surface area of ~2500 m^2^/g, and a pore volume of ~11 cm^3^/g were formed via annealing of colloidal silica with sucrose at 1000 °C followed by removal of SiO_2_ before being activated by CO_2_ at 900 °C [85]. Honeycomb-like porous carbon nanosheets with a surface area of 2038 m^2^/g and pore volume of 1.07 cm^3^/g were synthesized from the coal tar pitch using melamine as a soft template coupled with KOH activation [86].

#### 4.1.4. Organic Soft Templates

Ionic copolymers (i.e., cetrimonium bromide, cetrimonium chloride, and Gemini-type) and non-ionic copolymers (i.e., Pluronic F127, polyvinylpyrrolidone, Brij) are the common organic templates for the production of HA-PCs. N-doped (4.16–6.74 %) hierarchical mesoporous carbon spheres with a tunable size (30–140 nm), great surface area (1215–1517 cm^2^ /g), large pore volume (1.12–3.22 cm^3^ /g), and open interconnected mesoporous structure (5–20 nm) were synthesized using Gemini surfactant pentane-1,5-bis(dimethylcetyl ammonium bromide) in ethanol via the sol-gel method based on the interfacial reaction mechanism (Figure 3b) [84]. The obtained nanoparticles had a spherical size of 95 nm with a good distribution of C (69.23%), N (3.29%), O (19.77%), and Si (7.86%) (Figure 3c–g). Changing the amount of Gemini surfactant substantially increased surface area and porosity, but the TEOS was also used to generate a SiO_2_ template that was etched by hydrofluoric acid.

N,O,S-enriched hierarchical porous carbon foam with a surface area of 2685 m^2^/g was made using 1,3,5-trimethyl benzene (TMB) and Poloxamer 407 (F127) as a soft template in the presence of graphene oxide, dopamine (DA), and cysteine through the freeze-drying and chemical etching method (Figure 4a) [87]. F127 and TMB are assembled on graphene oxide into micelles coated with polydopamine obtained from the self-polymerization of dopamine into the surface of spherical micelles driven by the shearing force. Regardless of the difficulty of this approach, an additional activation step by KOH is needed.

#### 4.1.5. Ionic Liquids

With their thermal durability, small vapor pressure, and inbuilt heteroatoms, ionic liquids are commonly used as the precursors and templates to prepare heteroatom-doped porous carbon nanostructures. N/B-doped porous carbon nanosheets with a surface area of 2000 m^2^/g and pore volume of 2.75 mL/g were prepared using Bmp-dca and Emim-tcb ionic liquids as carbon precursors and templates with eutectics as a porogen at 1400 °C [90]. Notably, porogen could be easily recovered for reusability, and the obtained surface area (2000 m^2^/g) of the resultant carbon is superior to zeolite, activated carbons, and graphene. Moreover, the salt templating method is feasible for the large-scale and sustainability requirements.

N/S-co-doped hierarchical porous carbon nanosheets with a surface area of 575 m^2^/g and pore volume of 0.55 m^3^/g were formed using the self-assembly and carbonization of [Phne][HSO_4_], a protic ionic liquid, as the N/C/S source in addition to acting as a template with and soft template of OP-10 and F-127 (Figure 4b,c) [88]. Changing the ratios of [Phne][HSO_4_]/F-127/OP-10 leads to changing the surface areas and pore volume; meanwhile, OP-10/F-127 is easily decomposed during annealing at 700 °C to produce porous carbon foam with a good distribution of C, N (3.41%), and S (6.65%) (Figure 4c) [88]. This process is applicable for other ionic liquids with and without soft templates for the formation of HA-PC nanostructures.

Three-dimensional porous carbon nanosheets with a surface area of 1593 m^2^ g and pore volume of 0.85 cm^3^/g were formed via coal tar using 1-butyl-3-methylimidazolium tetrafluoroborate (BMIMBF_4_) ionic liquid as a template at 800 °C and in situ KOH activation [91]. Essentially, coal tar pitch transforms into polynuclear aromatic polymers at initial heating and BMIMBF4 decomposes at 500 °C, creating 3D interconnected pores coherently distributed within obtained carbon. The activation pivots significantly in increments of the surface area and pore volume. The utilization of ionic liquid for synthesizing carbon nanostructures is not emphasized enough, and their mechanisms remain ambiguous and not profoundly investigated. In addition, the high cost, the difficulty of preparation, and the air-sensitive nature of ionic liquids are crucial barriers to their commercialization.

Deep eutectic solvents (DESs) are novel ionic liquids consisting of Lewis or Brønsted acids and alkaline eutectic mixture systems. Unlike traditional ionic liquids, they possess outstanding physiochemical merits such as ionic strength, polarity, supramolecular structure, dielectric constant, inferior vapor pressure, biodegradability, low cost, and durability without environmental sound, which enables their utilization as a carbon source, templates, and solvents in the synthesis of porous doped carbon nanostructures [92].

Notably, utilization of DESs in the fabrication of porous carbon nanostructures is rarely reported and not studied enough compared with other templates. N/O-enriched hierarchically nanoporous carbon with a surface area of 1414 m^2^/g and pore volume of ~0.55 cm^3^/g was synthesized through the direct annealing of DESs of urea and ZnCl_2_ with phenol-formaldehyde resin 85. This allowed a high content of N (8.09 at.%) and O (14.77 at.%) tunable surface area/pore volume via adjusting the composition of the DESs and their ratio to resin. N/O-doped hollow carbon nanorods with a surface area of 1086 m^2^/g were prepared by direct carbonization of DESs (urea, 2,5-dihydroxy-1,4-benzoquinone, and ZnCl_2_) as sources of N/O/C and as a template [93]. ZnCl_2_ can act as a dehydration agent and pore modulator as it can generate ZnO as a self-template for the generation of mesopores and macropores.

#### 4.1.6. MOF Template

MOFs act as sources of carbon and self-templates for various kinds of porous carbon nanostructures, owing to their unique properties such as tunable chemical compositions, porosity, and surface area [51]. During the thermal decomposition process, the metal species inside MOFs can be directly used as the template. N-doped hollow carbon nanofibers with a surface area of 443.5 m^2^/g and pore volume of 1.6 cm^3^/g were synthesized via annealing of zeolite imidazole framework (ZIF-8) nanoparticles into electrospun polyacrylonitrile (PAN) at 800 °C (Figure 4d) [89]. The electrospinning of PAN with ZIF-8 forms nanofibers that are carbonized into porous carbon fiber (Figure 4e) at high temperatures, while Zn in ZIF-8 can form ZnO that acts as a template, and increasing the annealing temperature leads to decreasing the surface area and pore volume. The N and O contents were about 9.39% and 4.94%, respectively, as confirmed by the elemental mapping analysis (Figure 4e). The derived BET and Langmuir surface areas were 1192 and 1678 m^2^/g, respectively, with a large pore volume of 1.06 cm^3^ g^−1^, which is high enough to facilitate ion transportation. Electrospinning can be used for spinning any other polymer with different MOFs to form 1D nanostructures (i.e., rods, fibers, wires) driven by various experimental parameters (i.e., applied potential and polymer concentrations). Porous carbon nanorods with a surface area of 1192 m^2^/g and pore volume of 1.06 cm^3^/g were synthesized by thermal K-MOF at 200, 450, and 800 °C. K-MOF generates K_2_O and carbonates at high temperatures, which act as in situ templates and activators. The same concept is feasible for Zn-MOF, Ni-MOF, and Co-MOF because they can also create metal-oxides that act as templates ad activators. We briefly emphasized MOF-derived porous carbon nanostructures because it was deeply discussed before in another review [51].

#### 4.1.7. Biomass-Derived Carbons

Porous carbon nanostructures with various 1D, 2D, and 3D morphologies can be easily prepared via the direct annealing of biomass wastes from plant wastes, animals, insects, and microbes, which are low-cost, biodegradable, earth-abundant, unique porous structures and inherent heteroatoms, producing carbon nanostructures with outstanding electrical conductivity, massive active sites, and unique physicochemical properties. Moreover, biomass with its hierarchal porous structures (i.e., rods, sheets, nanotubes) acts as a carbon source and template for generating various porous morphologies under ambient conditions. N,S co-doped porous carbon nanosheets with a surface area of 1533 m^2^/g and pore volume of 0.92 cm^3^/g were synthesized via the one-step thermal decomposition at 400 °C and KOH activation at 850 °C of the biomass of willow catkin fibers [94].

B/N co-doped carbon nanosheets with a surface area of 416 m^2^/g and a pore volume of 0.76 cm^3^/g were created via annealing of gelatin biomass with boric acid at 900 °C [95]. The same approach is vulnerable to other biomasses. N-doped hierarchical porous carbon with a surface area of 315 m^2^/g and pore volume of 0.65 cm^3^/g was prepared from Bohai shrimp shell with KOH and CaCO_3_ at 700 °C [96]. CaCO_3_ acts as a self-template, while activation with KOH creates abundant micropores and mesopores [96]. The removal of CaCO_3_ significantly affected surface area and porosity. Soft templates and activators can be used with biomass to create porous carbon nanostructures. Moreover, selecting biomass with abundant heteroatoms is essential to allow in situ generation of heteroatom-enriched porous carbon. Notably, various plant wastes have not yet been investigated for the formation of HA-PCs. Meanwhile, previously reported biomass-derived porous carbon nanostructures have rarely been studied for CO_2_RR compared with other applications such as supercapacitors.

### 4.2. CO_2_RR Pathways

CO_2_ is a valuable molecule, and CO_2_RR can generate wide ranges of high value-added hydrocarbons and gases (i.e., CO and CH_4_, CH_3_OH, C_2_H_5_OH, and HCOOH) driven by the applied potentials (Table 2) and electron transfer number (Figure 1). This is based on the catalyst compositions and experimental parameters (i.e., applied potential, cell type, and electrolyte) via different CO_2_RR pathways. The CO_2_RR pathways on carbon-based catalysts generally include multiple interactive steps of CO_2_ adsorption and charge transfer, including electron and proton transfer in addition to product dissociation, which comprises migration of the products from the catalyst surface. The CO_2_ molecule is very stable thermodynamically with a great dissociation energy of ~750 kJ/mol for C=O (which needs significantly greater energy), which is greater than that of other carbon-based chemical bonds, such as C–H (~430 kJ/mol) and C–C (~336 kJ/mol). Notably, the applied potential in CO_2_RR is often much more negative than the equilibrium potential, resulting in high overpotential. This is because the CO_2_ adsorption comprises the rearrangement of a linear CO_2_ molecule to a bent radical anion via one-electron transfer to a CO_2_ molecule (i.e., CO_2_ + e^−^ → CO_2_^•−^, E^0^ = −1.9 V). During this process, the C=O bond is strongly unstable, and the electrons are shared between the CO_2_ and the catalyst. The obtained CO_2_^•−^ radical is highly reactive and easily reacts with H_2_O in the electrolyte to produce HCO_2_^•^ that is perturbed and tends to transform to HCO*, which is consequently released from the active catalyst sites. These multiple electron transfers and protonation processes occur on carbon-based catalysts to produce various products. The *E*^0^ for most CO_2_RR half-reactions is close to 0 V vs. RHE (Table 2), so the undesired side products because of the HER (2H^+^ + 2e^−^ → H_2_, *E*^0^, 0.42 V) result in the reduced electrocatalytic CO_2_RR activity of carbon-based catalysts. The CO_2_RR pathways can lead to the formation of various products driven by electron transfer and protonation under different applied potentials (Figure 1). Experimental studies using in situ analysis and computational calculations (i.e., density functional theory (DFT)) were used to confirm the CO_2_RR mechanisms and pathways. Mainly after the CO_2_ activation, protonation can occur for O to form *COOH or C to produce *OCHO. HA-PC catalysts *COOH pathways are preferred over *OCHO, so they usually produce CO in addition to HCOOH via the two-electron reduction process with rare CH_3_CH_2_OH. This is due to the absence of stable adsorbed states for the *HCOOH and *CO on the heteroatom-doped carbon surface for facilitating protonation of the *COOH intermediate. Despite the great progress in CO_2_RR, its mechanism and the boundary between each reaction pathway remain ambiguous owing to the absence of effective methods to detect the intermediates.

### 4.3. Heteroatom-Doping Configuration and Effects

#### 4.3.1. Advantages of Heteroatoms

There are various metal-based catalysts for CO_2_RR; however, their high cost, earth scarcity, and ability for the HER are critical barriers to large-scale applications and sustainability requirements. To this end, noble metals (i.e., Pt, Pd, Ru, and Au) possess great activity for the HER, while Au, Ag, and Sn, for example, produce mainly C1 (e.g., CO and HCOOH) rather than C2 products via a two-electron transfer pathway. Cu/Cu–O forms low-carbon hydrocarbons and oxygenates (i.e., CO with relatively high overpotentials). HA-PCs are promising electrocatalysts for the CO_2_RR with remarkable catalytic activity, long durability, and high selectivity. Distinct from metal-based catalysts, porous carbon materials possess numerous advantages such as tailorable structures/properties, rich specific surface chemistry, excellent surface area to volume ratio, thermal–chemical–physical durability, and low toxicity.

Moreover, C can be easily synthesized in high yield from inexpensive and natural abundant resources that meet the sustainability requirements. During CO_2_RR, HA-PCs tune the adsorption of CO_2_, induce electrolyte–electrode interaction, provide massive catalytic active sites, maximize atomic utilization, and tolerate the adsorption of intermediates and products. The HER activity of porous carbon catalysts is inferior, so the undesired effect of HER during CO_2_RR is neglected. The CO_2_RR is carried out in aqueous electrolyte solutions, so the electrode’s wettability or hydrophilicity is crucial to controlling the transportation of hydrated CO_2_ to the active sites. Heteroatom dopants enhance the hydrophilicity of the C-electrode, which can promote the accessibility of reactants to the active sites; make active sites more accessible, thus maximizing atomic utilization during the CO_2_RR. Integration of heteroatoms (i.e., P, N, O, B, and S) into carbon skeleton structures is essential to detect the limitation of electrical conductivity of carbon-based catalysts to empower their CO_2_RR activity. This is due to electron-donating or electron-withdrawing properties of heteroatoms that modulate the electronic characteristics.

#### 4.3.2. Nitrogen (N) Configuration and Effects

The N atom is an ideal heteroatom for modulating the properties of carbon due to its size (155 pm) being close to that of the carbon atom (170 pm). In addition, N has a larger electronegativity (3.04) than carbon (2.55), so N-doped C (N–C) can attract electrons, create multiple active sites, and augment the electronic/ionic conductivity of carbon. The N atom in the C skeleton structure is mainly pyridinic (398.5 eV), pyrrolic/pyridonic (400.1 eV), quaternary, or graphitic (401.1 eV), and pyridine-N-oxide (403.2 eV), that can be easily identified by X-ray photoelectron spectroscopy (XPS) [99]. These N species (Figure 2a) are highly active sites for CO_2_RR and other catalytic applications. N-doping introduces Lewis basicity on C’s surface that endows the CO_2_ adsorption significantly.

#### 4.3.3. Boron (B) Configuration and Effects

The smaller size of the B atom (85 pm) and its lower electronegativity (2.04) than C lead to the retention of the planar structure of C. The XPS showed two in-plane binding structures in B-doped C, including graphitic B at 200.5 eV, B atoms substituting for C atoms in the hexagonal ring, and “boron silane” B at 198.5 eV, referring to the place of B in the co-conjugated system (Figure 2b) [99]. B-doping persuades charge polarization in the C framework, stabilizes the negatively polarized O atoms in CO_2_, and consequently enhances the chemisorption of CO_2_ on C during the CO_2_RR process due to the relatively large electropositivity between B and C atoms.

#### 4.3.4. Sulfur (S) Configuration and Effects

Stone–Wales defects ease the formation of S-doping into carbon. The relatively larger size of the S atom (180 pm) and higher electronegativity (2.58) than C lead to improving C’s electrical conductivity and provide a higher spin density, edge strain, and charge delocalization. There are four types of S dopant in C, including S1, S2/S3, S4/S5, and S6 for adsorption of S on the C surface, the substitution of C by S at the edges, formation of the S/S oxide at the edges, and S-containing ring connecting sheets, respectively, as confirmed by DFT on a graphene model (Figure 2c) [100]. The formation energies for these species are different, but S1 is the most durable structure.

#### 4.3.5. Phosphorus (P) Configuration and Effects

The P atom has a lower electronegativity (2.19) and greater electron-donating ability than the C atom and generates positive charges on the P dopant and negative charges on positively charged C, introducing new functional groups (P^+^–C^−^) which change the structure of C, resulting in accelerated charge transfer and the reduced binding energy of intermediates. Also, the P atom, with its size (195 pm) being larger than C, leads to various defects in C skeleton structures, which act as active sites during CO_2_RR. Usually, P-doping occurs through the substitution of C by P. The XPS can detect P dopant at 130.2 eV, P–C at (132.5 eV), and P–O at (134.0 eV).

#### 4.3.6. Oxygen (O) Configuration and Effects

The oxygen (O) atom has a lower atom size (152 pm) than C but a greater electronegativity (3.44 pm) that enhances positive charges on C to form (C^+^–O^−^) after doping, which can attract electrons, create multiple active sites, and enhance the electronic/ionic conductivity of C. Also, introducing O into the carbon framework creates abundant structural defects and modulates the inherent electronic structure, allowing tuning adsorption/desorption of reactants and intermediates during CO_2_RR. Moreover, O-doping creates various functional groups (i.e., C–O, C=O, and C–OH) which can promptly activate oxidants allowing in situ generations of reactive oxygen species, activating H_2_O_2_ to in situ form oxygenated species that facilitate the CO_2_RR kinetics. The O-doping occurs via partial substitution of C with O to form C–O at (532 eV) and C=O at 533 eV species, as confirmed by the XPS, which are highly active sites for CO_2_RR. The FTIR can also easily detect the O-doping.

### 4.4. Single Heteroatom-Doped Porous Carbon Materials

Because the stable sp^2^ or sp^3^ hybridized carbon atoms cannot activate CO_2_ molecules, pristine carbon materials possess poor electrocatalytic activity for CO_2_RR. Therefore, incorporating heteroatoms with different electronegativity into carbon provides a great opportunity to modulate the electronic properties and thus improve their catalytic performance [50,102]. Porous carbons have unique advantages such as high specific surface area and tunable pore structure, which is beneficial for mass transfer and abundant active sites, thereby increasing the catalytic activity [103,104,105]. According to the reported literature, the frequently used single dopant can be classified into nitrogen, boron, sulfur, and fluorine [33]. Table 3 shows a detailed comparison of the electrocatalytic performance of binary HA-PCs toward CO_2_RR.

#### 4.4.1. Nitrogen-Doped Porous Carbon Materials

Nitrogen is one of the most widely studied dopants for carbon materials, based on the fact that the nitrogen atom has a similar atomic size to the carbon atom and higher electronegativity of 3.04 compared to the carbon atom’s 2.55. N-doping can induce more electrons into the carbon matrix and thus improve the electrical conductivity. Moreover, more active sites can be created after the N-doping [106,107,108]. As displayed in Figure 5A, the N dopants can exist in four types: pyridinic N, pyrrolic N, graphitic N, and oxidized N [109]. Liu et al. developed a series of N-doped carbon catalysts with tunable types and contents of nitrogen dopants to uncover the correlation between N species and catalytic performance toward CO_2_RR [110]. Electrochemical tests coupled with X-ray photoelectron spectroscopy identified that the CO_2_RR activity is proportional to the content of pyridinic N, whereas no noticeable relevance was observed on other N species. Moreover, the free energy calculated by density functional theory (DFT) calculations in Figure 5B revealed that the COOH could interact with pyridinic N in optimal bonding strength, benefiting the reduction of CO_2_ to form *COOH, and further to *CO. Apart from the pyridinic N content, porous carbon materials’ pore structure also affects the electrochemical performance [106,109,111,112,113,114,115]. For instance, N-doped nanoporous carbon sheets were synthesized by Yao et al. via a hydrothermal reaction and calcining Typha in NH_3_ [116]. It was found that the calcination temperature has a significant effect on the pore structure and N atom type. The optimal sample possesses the highest surface areas and pore volume, exposing abundant accessible pyridinic N, thereby delivering a much higher selectivity of 90% for CO at a lower overpotential of −0.31 V. However, there remains controversy about the critical active N sites for CO_2_RR [117]. For example, Liu and co-workers verified that the stable graphitic nitrogen atoms restricted in the micropores for coal-based metal-free electrocatalysts could effectively convert CO_2_ into CO [118]. Huang’s group demonstrated that pyridinic and graphitic N are the active sites for CO_2_RR by calcinating oxygen-rich Zn-MOF-74 precursors at different temperatures [119].

Up to now, the reported products for N-doped porous carbon materials toward CO_2_RR mainly focus on C1 compounds such as CO, CH_4_, and HCOOH [121,122]. It remains a challenge to stabilize the active C1 intermediates for their coupling to generate multi-carbon products. It is worth mentioning that Song et al. developed a nitrogen-doped ordered cylindrical mesoporous carbon (denoted as c-NC) as a high-efficient catalyst for the electroreduction of CO_2_ to ethanol with nearly 100% selectivity [120]. To reveal the unique structural effect of c-NC, an inverse mesoporous N-doped carbon with a similar pore structure and N content, namely i-NC, was synthesized as a control. It can be seen from Figure 5C that both c-NC and i-NC electrodes exhibit an obvious reduction peak in CO_2_ saturated 0.1 M KHCO_3_ solution, whereas no reduction peak was observed in Ar, indicating that the CO_2_RR occurs on c-NC and i-NC. Figure 5D shows the Faradaic efficiency (FE) of CO_2_RR on c-NC and i-NC catalysts at a potential between −0.40 and −1.00 V. The main product for c-NC and i-NC catalysts is ethanol from CO_2_RR and H_2_ from the hydrogen evolution reaction (HER), accompanied by a small amount of byproduct CO. This demonstrates that the mesoporous structure of both c-NC and i-NC contributes to the generation of ethanol. For c-NC, ethanol is the dominant product at the potential between −0.40 and −0.90 V, and the ethanol FE attained the maximum of 77% at −0.56 V. Meanwhile, the competitive reaction over c-NC for CO_2_ electroreduction to CO can be neglected. By contrast, the CO_2_ electroreduction to ethanol over i-NC is only dominated at the potential of −0.40 and −0.50 V, and the maximum ethanol FE reaches 44% at −0.50 V. After that, the HER becomes dominant, accompanied by a substantial amount of CO. Electrochemical impedance spectroscopy (EIS) was measured at −0.56 V vs. RHE to determine the charge transfer resistance (R_ct_) on c-NC and i-NC catalysts (Figure 5E). The smaller R_ct_ value for c-NC (3.8 Ω) than that for i-NC (8.5 Ω)) implies the easier transportation of electrons to the cylindrical surface of c-NC. DFT calculations were performed to check the possible CO_2_RR reaction pathways. Both pyridinic and pyrrolic N sites can promote the adsorption/activation of CO_2_ molecules and generate CO* intermediates (Figure 5F). The calculated reaction energy for CO* formation on pyridinic and pyrrolic N sites is −1.68 and 0.12 eV, respectively, implying the preferential formation of CO* on pyridinic N sites. The CO* intermediates can be stabilized by the cylindrical surface of c-NC with high electron density, thereby restricting the CO generation. The dimerization of CO* intermediates can proceed to form OC–CO* intermediates with reaction energies of −1.31 and −0.34 eV on pyridinic and pyrrolic N sites, respectively, signifying the favorable C–C bond formation on pyridinic N sites.

Moreover, the electron-rich cylindrical surface can accelerate the subsequent single and multiple proton-electron transfers to form the OC–COH* intermediate and ethanol. In short, the cylindrical surface with a high electron density and highly active N sites for c-NC synergistically benefits the highly efficient and highly selective production of ethanol. In addition, Yuan and co-workers reported the use of nitrogen-doped porous biochar from plant moss to catalyze CO_2_RR into CH_4_, CH_3_OH, and CH_3_CH_2_OH at a high current density and low overpotential [123]. Except for the above-mentioned C2 product, Li et al. first revealed that C3 hydrocarbons could be formed during CO_2_RR when nitrogen sites are situated close to each other in the micropore space [124]. Though great progress has been made for N-doped porous carbon materials, N sites’ high spin density also benefits the competitive HER, resulting in moderate FE and low partial current density.

#### 4.4.2. Other Metal-Free Heteroatom (S, F, or B)-Doped Porous Carbon Materials

Similar to N atoms, the doping of F and S atoms with electro-withdrawing or B atoms with electron-donating behaviors can also tailor the electronic structure of adjacent carbon atoms, thus improving the electrocatalytic performance toward CO_2_RR [125].

Nitrogen-doped carbon nanotubes (NCNTs) with an average diameter of 30 nm and N content of 5.0% were formed via the liquid chemical vapor deposition (CVD) method using acetonitrile and dicyandiamide with ferrocene at 850 °C under Ar/H_2_. NCNTs promoted the CO_2_RR with a maximum FE of CO of nearly 80% at −0.78 V and an overpotential of −0.26 V, which is comparable to Au and Ag nanoparticles but with a lower overpotential in 0.1 M KHCO_3_ [126]. This was significantly higher than that of N-free CNTs revealing inferior CO_2_RR activity (3.5% FE_CO_). NCNTs remain stable at −0.8 V for 10 h without any obvious degradation, and the FE varies slightly around 80% [126]. For instance, a fluorine-doped cage-like porous carbon (F-CPC) electrocatalyst was synthesized by a polymer-derived method using SiO_2_ spheres as templates (Figure 6A) [127]. The transmission electron microscope (TEM) image in Figure 6B clearly shows the hollow cage-like structure of F-CPC with a thin carbon shell, in which the bright dots marked by yellow circles represent the rich mesopores on the surface. Elemental mapping images reveal that the carbon and fluorine elements homogeneously distribute throughout the F-CPC (Figure 6C,D). There is a single peak at 689.8 eV in the X-ray photoelectron spectroscopy (XPS) spectra of F 1s, suggesting the presence of a semi-ionic C–F bond in the F-CPC (Figure 6E). Compared with the covalent C–F bond, the semi-ionic C–F bond is anticipated to promote the CO_2_RR. The F-CPC catalyst exhibits a maximum CO FE of 88.3% at −1.0 V vs. RHE among all samples (Figure 6F), originating from the novel structure and morphology. Moreover, the superior stability of F-CPC was confirmed by chronoamperometric curves at −0.9 V vs. RHE (Figure 6G).

**Table 3 nanomaterials-12-02379-t003:** Comparison of electrocatalytic performance of mono HA-PCs toward CO_2_RR.

Electrocatalysts	Synthetic Method	Electrolyte	Main Product	Potential of FE_max_	sFE_max_ (%)/*j*_CO_ (mAcm^−2^)	Durability	Refs.
				(vs. RHE)			
NC-900	Hydrothermal synthesis and calcination of Typha in NH_3_ at 900 °C	0.5 M KHCO_3_	CO	−0.5	82%/~1.25 mA·cm^−2^	FE_CO_ stability 75% after 10 h	[116]
N-GRW (GM2)	The first polymerization of melamine and L-cysteine to form C_3_N_4_ at 600 °C, followed by carbonization at higher temperatures	0.5 M KHCO_3_	CO	−0.4	87.6%/~7.8 mA·cm^−2^	FE_CO_ stability 80% after 16 h	[110]
TTF-1	Thermal treatment of 2, 6- dicyanopyridine and ZnCl_2_ at 600 °C for 40 h	0.5 M KHCO_3_	CO	−0.68	82%/~−1 mA·cm^−2^	FE_CO_ stability 75% after 12 h	[115]
c-NC	A soft-template method via the self-assembly of resol, F127, and dicyandiamide	0.1 M KHCO_3_	CH_3_CH_2_OH	−0.56	77%/~−0.35 mA·cm^−2^	FE_CO_ stability 77% after 6 h	[120]
MNC-D	Pyrolysis of ZIF-8 at 900 °C for 3 h and mixed with HCl, followed by treatment in dimethylformamide	0.1 M KHCO_3_	CO	−0.58	∼ 92%/∼−6.1 mA·cm^−2^	FE_CO_ stability ∼86% after 16 h	[108]
NPC-1000	High-temperature annealing of the mixture of oxygen-rich Zn-MOF-74 and melamine at 1000 °C	0.5 M KHCO_3_	CO	−0.55	98.4%/−3.01 mA·cm^−2^	FE_CO_ stability ∼98% after 21 h	[119]
NPC-900	One-step pyrolysis method via the self-assembly of anthracite coal, KOH, and dicyandiamide	0.5 M KHCO_3_	CO	−0.67	95%/−4.8 mA cm^−2^	FE_CO_ stability ∼ 80% after 10 h	[118]
BAX-M-950	Soaking commercial activated carbon BAX-1500 in a melamine suspension in ethanol followed by evaporation and drying, then heating at 950 °C in N_2_	0.1 M KHCO_3_	CO, CH_4_	−0.66, −0.76	40%, 1.2%/−3 mA cm^−2^	FE_CO_ stability ∼20%, 1.1% after 24 h	[111]
CNPC-1100	Etching coal powder in ammonia atmosphere	0.1 M KHCO_3_	CO	−0.6	92%/−4.6mA cm^−2^	FE_CO_ stability ∼62.5% after 8 h	[112]
WNCNs-1000	An NH_3_ etching strategy by using NaCl and coal tar pitch as templates and precursor	0.1 M KHCO_3_	CO	−0.49 (overpotential)	84%/~−1.26 mA cm^−2^	FE_CO_ stability ∼81% after 8 h	[106]
NDC-700	One-step pyrolysis of wheat flour and KOH	0.5 M NaHCO_3_	CO	−0.82	83.7%/~−8 mA cm^−2^	FE_CO_ stability ∼79.4% after 2 h	[109]
PNC	High-temperature calcination by using melamine as the nitrogen source and pentaerythritol as the carbon source	0.1 M KHCO_3_	CO	−0.6	74%/~−4 mA cm^−2^	FE_CO_ stability ∼70% after 10 h	[121]
N/C-Cl-1100	Halogen-assisted calcination of ZIF-8 at 1100 °C	0.1 M KHCO_3_	CO	−0.5	99.5%/~−2.6mA cm^−2^	FE_CO_ stability ∼99% after 20 h	[117]
HPC	Hydrothermal treatment of moss at 180 °C for 24 h, followed by pyrolyzing at 900 °C for 2 h and acidic etching	0.5 M KHCO_3_	CH_4_, C_2_H_5_OH, CH_3_OH	−1.2 (vs. Ag/AgCl)	56, 26, 10.5%/~−15 mA cm^−2^	FE_CO_ stability ∼92.6% after 30 h	[123]
NDAPC	Pyrolysis of petroleum pitch under nitrogen atmosphere followed by ammonia etching	0.1 M KHCO_3_	CO	−0.9	83%/~−3.76 mA cm^−2^	FE_CO_ stability ∼80% after 8 h	[113]
NG-800	The first formation of 3D graphene foam by chemical vapor deposition and post-doped with graphitic-C_3_N_4_, followed by etching Ni with HCl	0.1 M KHCO_3_	CO	−0.58	85%/~−1.8 mA cm^−2^	FE_CO_ stability ∼80% after 5 h	[114]
NPC-600	Hydrothermal treatment of SBA-15 and digested sludge	0.1 M NaHCO_3_	Formate	−1.5 (vs. SCE)	68%/~−7.5 mA cm^−2^	FE_CO_ stability ∼68% after 4 h	[122]
P-NC	The calcination of sucrose, urea, and NaCl at 800 °C for 4 h	0.5 M KHCO_3_	CO	−0.8	81.3%/~−7.2 mA cm^−2^	FE_CO_ stability ∼81% after 6 h	[104]
NC1100	The calcination of ZIF-8 at 1100°C in Ar	0.5 M KHCO_3_	CO	−0.5	95.4%/~−3 mA cm^−2^	FE_CO_ stability ∼90% after 20 h	[107]
F-CPC	An aldol reaction conducted at SiO_2_ surface, followed by calcination at 900℃, activation with CO_2_, and removal with HF	0.5 M KHCO_3_	CO	−1.0	88.3%/~−37.5 mA cm^−2^	FE_CO_ stability ∼85% after 12 h	[127]
FC	Pyrolyzing the mixture of commercial BP 2000 and polytetrafluoroethylene	0.1 M NaClO_4_	CO	−0.62	89.6%/~−0.25 mA cm^−2^	-	[128]
BG	Heating the uniform mixture of graphene oxide and boric acid at 900 °C in Ar	0.1 M KHCO_3_	HCOOH	−1.4 (vs. SCE)	66%/~−3 mA cm^−2^	FE_CO_ stability ∼66% after 4 h	[129]

Of note, F-CPC can maintain 97% of the initial current density and keep CO FE stable for 12 h. It turns out that the nanocage structure of F-CPC can generate an enhanced electrostatic field and increase the K^+^ ion concentration, thereby lowering the thermodynamic energy barrier for CO_2_RR. In another report [128], a fluorine interlayer doped carbon (FC) catalyst was obtained by the facile pyrolysis of the precursor’s mixture, and the FC was able reach the maximum FE for CO of 89.6% at −0.62 V. DFT calculations revealed that fluorine interlayer doping can activate neighbor carbon atom defects and contribute to the interaction of COOH* with activated carbon, which is considered as the rate-determining step for CO_2_-to-CO conversion. Likewise, S-doping can also result in higher spin density and charge delocalization owing to the dissimilar electronegativity of sulfur (2.58) and carbon (2.55), which are believed to enhance the electrocatalytic activity for CO_2_RR. S-doped porous carbon nanosheets (CPSs) were prepared by the annealing of poly (4-styrene sulfonic acid-co-maleic acid) sodium salt at 800 °C under N_2_, which showed a higher CO_2_RR current density (7.2 mA/cm^−2^) than that of N-doped CPSs (CPSNs) (7.2 mA/cm^−2^) [130]. The maximum FE for CO (FE_CO_) and CH_4_ on the CPSs was about (2.0 % and 0.1%) at −0.99 V vs. RHE relative to the CPSNs (11.3% and 0.18%) due to S and/or N dopants, which stabilize the CO_2_^−^ and COOH* intermediates that promote CO2RR to CO and CH_4_. The durability studies showed that the CPSs maintained only 65% of their FE_CO_ (1.3%) after 2 h, while the CPSNs kept around 72.7% of their FECO (8%). With the relatively large difference between the electronegativity of B and C atoms, B-doping can induce charge polarization and make the carbon framework suitable for adsorbing CO_2_ molecules. For instance, B-doped graphene (BG) was synthesized by catalyst by heating the graphene oxide and boric acid (1/5 wt ratio) at 900 °C under Ar, which showed higher CO2RR activity with a current density of ~6 mA/cm^2^ at 1.6 V than undoped graphene (~1.6 mA/cm^2^), and Bi (~2.4 mA/cm^2^) in 0.1 M KHCO_3_ can reach an FE for formic acid of 66% at −1.4 V vs. SCE during CO_2_RR [129]. BG mainly allowed the CO_2_RR to formate with an FE of 66% at −1.4 V, which was substantially higher than Bi (FE of 20%). BG remains stable for 4 h without any significant loss in the CO_2_RR activity. B-doped diamond (BDD) thin films were grown on Si(111) wafers by the microwave plasma-assisted chemical vapor deposition (MPVCD) method at 5 kW using B(OCH3) (B-source) and acetone (C-source) with B/C (1/1 atomic ratio) [131]. BDD showed a current density of 0.3 mA/cm^2^ at 2.2 V vs. Ag/AgCl in an electrolyte of methanol solution and tetrabutylammonium perchlorate (MeOH-TBAP). Moreover, BDD allowed the CO_2_RR to produce formaldehyde, formic acid, and H_2_ with a maximum FE of 74% at −1.7 V, 15% at −1.5 V, and 1.1% at <−1.7 V, respectively [131]. Interestingly, BDD revealed a higher CO_2_RR activity with a greater FE for formaldehyde, formic acid, and H_2_ in MeOH-TBAP electrolyte relative to water (0.1 M NaCl) and seawater, respectively. Notably, mono heteroatom-doped carbon-based catalysts for CO_2_RR are rarely reported and are not studied enough (Table 3). Also, other reports did not systemically study the activity and durability as a function of dopant amount or in different electrolytes.

### 4.5. Binary Heteroatom-Doped Porous Carbon Materials

In consideration of the unsatisfying catalytic performance for single heteroatom-doped porous carbon materials, dual heteroatom co-doping may regulate the chemical properties in a wide range and bring great promise toward CO_2_RR by virtue of the synergistic electronic interactions between different dopants. Table 4 shows a detailed comparison of the electrocatalytic performance of binary HA-PCs toward CO_2_RR.

#### 4.5.1. Nitrogen, Sulfur Co-Doped Porous Carbon Materials

Although there are few reports about the effect of sulfur dopants into porous carbon materials on CO_2_RR, the co-doping of nitrogen and sulfur has been extensively investigated over the past few years [40,43,130,132,133,134,135,136]. Yang et al. reported N, S co-doped hierarchically porous carbon nanofiber (NSHCF) membranes as high-efficiency catalysts for electrochemical conversion of CO_2_ to CO with 94% Faradaic efficiency at −0.7 V vs. RHE [133]. In their synthesis, NSHCF900 was obtained by electrospinning the mixture of ZIF-8 nanoparticles, trithiocyanuric acid (TA), and polyacrylonitrile (PAN), followed by the carbonization at 900 °C in Ar (Figure 7A). In contrast, the NHCF900 without sulfur doping can only achieve a CO FE of 63% toward CO_2_RR, highlighting the key role of S species in promoting electrochemical activity. Moreover, DFT calculations showed that the Gibbs free energy of *COOH on pyridinic N adjacent to the carbon-bonded S atom is effectively decreased compared to that on pure pyridinic N atoms (Figure 7B). This is likely due to the greater spin density and charge delocalization arising from S atom doping. Following this method, Li and coworkers synthesized N, S co-doped hierarchically porous carbon (NSHPC) by pyrolysis of glucosamine hydrochloride and thiocyanuric acid precursor using SiO_2_ as a hard template, and they were able to obtain a maximum CO FE of 87.8% at −0.6 V vs. RHE [134]. The N, S co-doped high-surface-area carbon materials (SZ-HCN) were developed by one-step pyrolysis of N-containing polymer and S powder [136]. A partial current density of 5.2 mA/cm^2^ at the overpotential of 0.490 V and a maximum CO FE of 93% at −0.6 V for CO_2_RR was achieved on SZ-HCN, which is superior to those on the single N-doped carbon counterpart.

In another study [40], Pan et al. concluded that sulfur addition could significantly boost the electrochemical activity and selectivity for CO_2_RR of N-doped carbon catalysts. A layer-structured carbon nitride-templated pyrolysis strategy synthesized the N, S dual-doped carbon (NS-C) layers. Thiourea and citric acid were used as N, S sources and C sources, respectively. For comparison, the NS-C samples (denoted as NS-C-800, NS-C-900, and NS-C-1000) were prepared at different annealing temperatures of 800, 900, and 1000 ℃ to tune the doped N, S content, respectively. In addition, S-free N-C (denoted as N-C-900) layers were also obtained under identical synthesis conditions to those of NS-C-900, except for replacing thiourea with urea. It is notable from Figure 7C that the S-doping is beneficial for forming highly active pyridinic N while suppressing graphitic N. The CO FE for each catalyst at various potentials is shown in Figure 7D. NS-C-900 can catalyze CO_2_RR at a smaller overpotential but with a larger CO FE relative to N-C-900, showcasing the improved reactivity and selectivity after sulfur doping.

Furthermore, it was found that the catalytic activity of NS-C layers strongly depends on the annealing temperature, among which the NS-C annealed at 900 °C can yield the highest CO FE of 92%. This is mainly attributed to doped N and S species’ optimal content and structure, which can expose more active sites and accelerate mass transport. DFT calculations were further employed to study the effect of sulfur doping on the inherent activity of various N dopants. They found that the distance between the S atom and pyridinic N is proportionally related to the enhancing effect of sulfur atoms. It was speculated that introducing S atoms can increase the spin density of N-C, which is more conducive to promoting electron transfer and COOH* adsorption, resulting in superior catalytic ability at a lower overpotential. Similarly, Li et al. revealed the relationship between the dispersion of S and N groups in porous carbon with the CO FE, verifying the enhancing effect of sulfur doping on highly active pyridinic N sites for CO_2_RR [132].

#### 4.5.2. Nitrogen, Phosphorus Co-Doped Porous Carbon Materials

In view of the larger difference between heteroatom P and N in electronegativity, dual-doping into porous carbon materials provides more options to boost the electrocatalytic performance toward CO_2_RR [137,138,139,140]. Chen et al. fabricated N, P co-doped carbon materials (NPCM-1000) using aniline monomer and phytic acid as nitrogen, carbon, and phosphorus sources via one-pot pyrolysis at 1000 °C (Figure 8A). As a reference, single N-doped carbon materials (NCM-1000) were synthesized by replacing phytic acid with HCl. The NPCM-1000 exhibits a similar structure to NCM-1000 but with a higher surface area, pyridine N content, and defects, as revealed by TEM, N_2_ adsorption–desorption measurement, X-ray photoelectron spectroscopy, and Raman spectra, respectively. The electrochemical tests were conducted using a standard three-electrode system using a CO_2_-saturated 0.5 M NaHCO_3_ solution. As expected, the NPCM-1000 exhibits a higher onset potential (−0.38 V) than NCM-1000 (−0.59) V (Figure 8B). The CO FEs at various potentials of NPCM-1000 and NCM-1000 are displayed in Figure 8C, in which the maximum FE for NPCM-1000 and NCM-1000 is 92% and 14% at −0.55 V, respectively. Moreover, NPCM-1000 shows much higher partial current densities of CO (*j_co_*) than NCM-1000 (Figure 8D). The Tafel curve is generally used to describe the reaction kinetics of CO_2_RR. The Tafel slope of 122 mV/dec for NPCM-1000 implies the formation of COOH* is the rate-determining step, while for NCM-1000 it is the CO_2_ molecular adsorption and desorption (Figure 8E). Based on DFT calculations, the N, P co-doping can synergistically promote the formation of COOH*. Furthermore, the reaction barrier of CO_2_ activation can be more effectively reduced for NPCM-1000 than that for NCM-1000, as calculated from Figure 8F,G. Recently, Liang and co-workers demonstrated a porous N, P dual-doped carbon nanosheet catalyst for CO_2_RR which can attain a high CO FE of 88% and good stability for 27 h at a low overpotential [140]. In addition, they found that the introduction of P can adjust the electronic structure of pyridinic N to hinder the adsorption of *H and contribute to the higher selectivity of CO_2_-to-CO.

#### 4.5.3. Nitrogen, Boron Co-Doped Porous Carbon Materials

Since N atoms have a larger electronegativity and B atoms have a smaller electronegativity than C atoms, N/B co-dopants may modulate the electronic structure and create an unexpected effect on CO_2_RR [33]. As reported previously [141], B, N co-doped nanodiamond (BND) was an efficient and stable electrocatalyst for CO_2_RR to ethanol. BND revealed a high FE of 93.2% at −1.0 V vs. RHE due to the synergistic effect of B and N co-dopants. Then, various HA-PCs co-doped with B and N were reported for the CO_2_RR [142,143]. Zhao’s group has reported the integrated design of a N, B co-doped three-dimensional hierarchical porous carbon network with a high doping level by a salt-sugar method [144].

The as-obtained catalyst displays a high CO_2_-to-CO FE of 83% at a low overpotential of 290 mV and good stability over 20 h. Based on the physical and electrochemical characterization, the superior activity and selectivity are first attributed to the unique porous structure, including macropores, mesopores, and micropores, which can offer larger surface areas and more active sites for CO_2_ adsorption. Moreover, the N-doping can accelerate the conversion of CO_2_ into the CO_2_* intermediate, while the B atoms can facilitate the capture of CO_2_ by bonding to the O atoms of the CO_2_* intermediate, and then the conversion of COOH* into CO*. B, N co-doped mesoporous carbon (BNMC) was synthesized through carbonization of the mixed precursors using glucose as the carbon source, urea and dicyandiamide as the nitrogen source, and boric acid as the boron source along with silica as a template (Figure 9A) [142]. The as-synthesized BNMC annealed at 1000 ℃ (BNMC-1000) possesses a porous surface, as seen from the scanning electron microscopy (SEM) image in Figure 9B. The TEM image of BNMC-1000 reveals the rich mesopores with an average diameter of 25 nm (Figure 9C). For comparison, control experiments of BNC-1000 without mesopores, NMC-1000 without B-doping, and BMC-1000 without N-doping were conducted to explore the effect of the porous structure and heteroatom doping. As shown in Figure 9D,E, the BNC-1000 reveals a very low current density and CO FE compared with NMC-1000 and BNMC-1000, indicating the critical role of the mesoporous structure in enhancing CO_2_RR. Furthermore, the current density and CO FE of BNMC-1000 is larger than those of NMC-1000, demonstrating that the N, B dual-doping contributes to improving CO_2_RR relative to single N-doping. They also confirmed that BMC-1000 could reduce CO_2_ to formic acid with a different electrochemical selectivity from NMC-1000 and BNMC-1000. The Tafel slope of BNMC-1000 was calculated to be 128 mV/dec, which is smaller than that of NMC-1000 (141 mV/dec), indicating the favorable reaction kinetics on BNMC-1000 through N, B dual-doping (Figure 9F). DFT calculations reveal that the coupling effect between N and B atoms can tune the projected density of states of adjacent carbon active sites, thereby generating an optimal adsorbed energy of *COOH and *CO on the carbon surface.

**Table 4 nanomaterials-12-02379-t004:** Comparison of electrocatalytic performance of binary and ternary HA-PCs toward CO_2_RR.

Electrocatalysts	Synthetic Method	Electrolyte	Main Products	Potential of FE_max_	FE_max_ (%)/*j*_CO_ (mAcm^−2^)	Durability	Refs.
				(vs. RHE)			
Binary HA-PCs							
CPSN	The carbonization of poly(4-styrenesulfonic acid-co-maleic acid) sodium salt at 800 °C, followed by impregnation with urea-saturated solution and holding at 800 °C in N_2_ for 30 min	0.1 M KHCO_3_	CO	−0.99	11.3, 0.18%/ ~−4 mA cm^−2^	FE_CO_ stability ∼8, 0.126% after 27, 2 h	[130]
			CH_4_				
NSHCF900	The carbonization of polymer nanofiber at 900 °C in Ar	0.1 M KHCO_3_	CO	−0.7	94%/ ~−103 mA cm^−2^	FE_CO_ stability ∼93% after 36 h	[133]
NS-C	The calcination of citric acid and thiourea at 550 °C for 2 h under Ar	0.1 M KHCO_3_	CO	0.49 (overpotential)	92%/ ~−2.63 mA cm^−2^	FE_CO_ stability ∼91% after 20 h	[40]
NS-CNSs-1000	Two-step pyrolysis of the mixture of iron-oleate, Na_2_SO_4_ and urea and acid etching	0.5 M KHCO_3_	CO	−0.55	85.4%/ ~−2.5 mA cm^−2^	FE_CO_ stability over 80% after 20 h	[43]
NSHPC	The pyrolysis of glucosamine hydrochloride and thiocyanuric acid precursor using SiO_2_ as hard templates	0.1 M KHCO_3_	CO	−0.6	87.8%/ ~−2.2 mA cm^−2^	FE_CO_ stability ∼80% after 10 h	[134]
SZ-HCN	One-step pyrolysis of N-containing polymer and S powder	0.1 M KHCO_3_	CO	−0.6	93%/ ~−5.2 mA cm^−2^	FE_CO_ stability ∼90% after 20 h	[136]
BAX-TU-20	High-temperature treatment of commercial wood-based carbon impregnated with thiourea	0.1 M KHCO_3_	CO	0.67	29, 0.27%/ ~−1.5 mA cm^−2^	FE_CO_ stability ∼22.5, 0.25% after 40, 50 h	[132]
			CH_4_				
NPC-900-2	Pyrolysis-controlled sacrificial templating approach using citric acid, melamine and NH_3_, and phytic acid as carbon, nitrogen, and phosphorous source, respectively	0.5 M KHCO_3_	CO	−0.41	88%/ ~−1.71 mA cm^−2^	FE_CO_ stability ∼80% after 27 h	[140]
NPCM-1000	One-pot synthesis by using aniline monomer and phytic acid as nitrogen, carbon, and phosphorus source	0.5 M NaHCO_3_	CO	−0.55	92%/ ~−1.25 mA cm^−2^	FE_CO_ stability ∼75% after 24 h	[139]
MPC-1000	Pyrolysis of vitamin B_12_ in NaCl assembly-enclosed nanoreactors	0.1 M KHCO_3_	CO	−0.7	62%/ ~−3.1 mA cm^−2^	FE_CO_ stability ∼60% after 20 h	[137]
N, P-FC	One-step soft-template pyrolysis method by using phytic acid as P source, dicyandiamide as N source, and polyethylene glycol as soft template	0.5 M NaHCO_3_	CO	−0.52	83.3%/ ~−8.52 mA cm^−2^	FE_CO_ stability ∼80% after 12.5 h	[138]
NBPC	Liquid nitrogen-assisted freeze-drying of the NaCl-glucose solution containing carbon, nitrogen, and boron precursors and two-stage solid pyrolysis	0.5 M KHCO_3_	CO	−0.4	83%/ ~−0.5 mA cm^−2^	FE_CO_ stability ∼80% after 20 h	[144]
BND3	The deposition of BND film on Si substrate using hot filament chemical vapor deposition method with a gas mixture of CH_4_/B_2_H_6_/N_2_/H_2_	0.1 M NaHCO_3_	CH_3_CH_2_OH	−1.0	93.2%/ ~−0.5 mA cm^−2^	FE_CO_ stability ∼93.2%) after 48 h	[141]
			CH_3_OH				
			HCOO^−^				
BNMC-1000	The carbonization of a precursor containing urea, dicyandiamide, glucose, and boric acid along with silica as templates	0.1 M KHCO_3_	CO	−0.55	95%/~−2.7 mA cm^−2^	FE_CO_ stability ∼90% after 10 h	[142]
Ternary HA-PCs							
NSP-HPC	A H_2_SO_4_-H_3_PO_4_ binary-acids activation method	0.5 M KHCO_3_	CO	−0.7, −1	92, 98.5%/~−5.2, −186 mA cm^−2^	FE_CO_ stability ∼91, 94% after 50 h	[145]
LC-3	The carbonization of the mixture of lignin, urea, melamine, NaCl, and ZnCl_2_ at 1000 °C for 2 h in Ar, followed by impregnating in HCl for 24 h	0.1 M KHCO_3_	CO	−0.6	95.9%/~−1.98 mA cm^−2^	FE_CO_ stability ∼95.9% after 18 h	[135]

### 4.6. Ternary Heteroatom-Doped Porous Carbon Materials

Introducing multiple heteroatoms with different electronegativities and sizes to porous carbon materials can modify the local electronic properties, thereby boosting the electrochemical CO_2_RR. Table 4 shows a detailed comparison of the electrocatalytic performance of ternary HA-PCs toward CO_2_RR. Yang et al. presented a facile synthesis of N, P, S ternary heteroatom-doped carbon (NSP-HPC) via the H_2_SO_4_-H_3_PO_4_ binary-acids activation method [145]. The dual heteroatom-doped NS-HPC, NP-HPC, and single-doped N-HPC were also prepared as a control. In a conventional H-cell, NSP-HPC exhibits the lowest onset potential of −0.38 V among the four samples, which corresponds to a smaller overpotential of 270 mV (Figure 9G). Moreover, NSP-HPC demonstrates the highest CO FE among the control samples at various potentials (Figure 9H). The maximum CO FE for NSP-HPC is 92%, achieved at −0.7 V, while that for NP-HPC, NS-HPC, and N-HPC is 83%, 90%, and 56%, respectively. More importantly, the current density and CO FE of NSP-HPC can keep constant for 50 h at −0.7 V, reflecting its robust stability. In situ Raman spectroscopy reveals that *COOH is the key intermediate in CO_2_-to-CO conversion. Based on the DFT calculations, the free energy diagrams of CO_2_RR for each catalyst are given in Figure 9I. During the CO_2_-to-CO conversion, the free energy barrier of *COOH is 1.61 eV for N-doped carbon. Once P or S atoms are introduced, the free energy barrier decreases to 1.42 and 0.54 eV, respectively. The N, S, and P doping reveals the lowest *COOH formation energy of 0.05 eV, indicating that the synergistic coupling effect among multiple heteroatoms could benefit the enhanced CO_2_RR. Furthermore, the well-defined hierarchically porous structure of NSP-HPC also plays a key role in improving the CO_2_RR activity.

## 5. Conclusions and Outlook

In summary, this review discussed the controlled synthesis and emphasized the rational synthesis of heteroatom (i.e., N, S, O, F, or B)-doped porous carbon nanostructures (HA-PCs) for the CO_2_RR. This includes the CO_2_RR fundamental pathways and engineering methods of HA-PC nanostructures. The effects of dopants (individually or mixed) on the CO_2_RR activity and durability were reported.

The Faradaic efficiency (FE), overpotential (*ƞ*), partial current density (j), durability, energy efficiency (*E_eff_*), and turnover frequency (TOF) are the main factors that determine the CO_2_RR activity of HA-PCs.

There are a few methods for the rational synthesizing of HA-PC nanostructures: template-based, activation, element-doping, and direct annealing of biomass-based resources, which vary in their productivity for tailoring the porosity (i.e., pore-volume, pore order, and pore size), surface area, and structure of HA-PCs. Hard template methods (Zn-based, Mg-based, Ca-based, and Si-based) are extensively studied for HA-PCs. This is due to their ability to produce uniform porous morphologies with well-controlled porosity and surface area, but their multiple reaction steps and use of hazardous chemicals for etching templates remain a grand challenge. However, the Zn-based template is the most preferred among hard templates because Zn is inexpensive, earth-abundant, and can be easily prepared in high yield and evaporate during annealing without additional etching steps. New templates such as melamine, dry ice, and MXene have also allowed the synthesis of HA-PCs, but they have rarely been reported. Soft templates including ionic/nonionic copolymers (i.e., F127and cetrimonium chloride) and ionic liquids (i.e., Gemini-type) are also used for the preparation of HA-PCs, but they cannot produce uniform porous structures with a high surface area and they are usually accompanied with other methods.

Deriving HA-PCs from biomass is among the most promising approaches due to the low-cost and natural abundance of biomass waste; also, the unique structures and composition of biomass wastes drive the production of HA-PCs with well-defined porosity and composition under ambient conditions that meet the sustainability requirements. Using melamine is highly promising as it allows integration of high N content and constructer carbon-nitride over HA-PCs during carbonization at a temperature above 500 °C. MOF-derived HA-PCs are also studied significantly, but their high cost and multiple/complicated reaction steps remain a significant challenge. Moreover, the use of activators is needed to enhance the surface area and porosity.

Various HA-PC nanostructures have been prepared using multiple methods for CO_2_RR, which varied in their performance (Table 3 and Table 4). Notably, KHCO_3_ and NaHCO_3_ are the main electrolytes used for HA-PCs, and NaClO_4_ electrolytes were rarely reported; furthermore, CO is the main CO_2_RR product, and other products such as CH_4_, HCOOH, C_2_H_5_OH, and CH_3_OH were rarely found, which implies the selectivity of HA-PCs for producing CO.

The biomass-derived HA-PCs are the most promising for practical applications. Chlorine-promoted N/S co-doped PCs (LC-3) formed from annealing of lignin, urea, melamine, NaCl, and ZnCl2 at 100 °C and impregnating in HCl revealed an FE of 95.9% [134] and N/B-doped porous carbon (BNMC-1000) showed an FE of 95% [142], and N-doped porous carbon from calcination of ZIF-8 (NC1100) showed an FE of 95.4% [107]. N-doped PCs are the most reported and most active compared to other HAs dopants, and they are the most active for CO_2_RR. The highest active HA-PCs are N-doped porous carbon (NPC-1000) formed using annealing of Zn-MOF-74 and with melamine at 1000 °C showing an FE of 98.4% [119] and N-doped porous carbon (N/C-Cl-1100) obtained from halogen-assisted annealing of ZIF-8 with KCl at 1100 °C revealing an FE of 99.5% [117]. This is due to their porosity and abundance of N-species (i.e., pyridinic-N and graphitic-N). B/N-doped nanodiamond (BND) produced only ethanol with an FE of 93.2% [141]. Despite the noticed progress in HA-PC catalysts for CO_2_RR, they are still far from being useful for large-scale applications, so various perspectives and challenges should be addressed:The current preparation approaches of heteroatom-doped porous carbon-based nanocatalysts involve multiple reaction steps, energy consumption, and hazardous reagents, making them impractical. Thus, they should be prepared using green materials under ambient conditions to meet sustainability requirements.Using biomass wastes is a promising approach to synthesizing HA-PCs with tunable porosity and surface area under ambient conditions; however, they are rarely reported for CO_2_RR.The CO_2_RR performance of HA-PCs is mainly measured in CO_3_-based electrolytes, so other organic, ionic liquid, and hybrid electrolytes should be studied to produce liquid products other than CO. Moreover, the effect of electrolytes and cell design on the CO_2_RR of HA-PCs has not yet been reported.Integration of HA-PCs with other materials such as carbon nitride [1,146,147,148], MXenes [23,149,150,151], carboxylated carbon/graphene [152,153], and graphdiyne [41] can enhance their CO_2_RR owing to their rich electron density, unique physicochemical properties, and catalytic/photocatalytic merits. Using HA-PCs with 3D porous multi-metallic nanocrystals (i.e., cages, branched, dendrites, and yolk-shell) can improve the CO_2_RR selectivity.Computational studies could be conducted with experimental studies to allow the synthesis of novel HA-PCs and to examine their CO_2_RR activity, mechanism, and pathways.

## Data Availability

The data presented in this study are available on request from the corresponding author.

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
