# Peer review of "Heteroatom-Doped Porous Carbon-Based Nanostructures for Electrochemical CO2 Reduction"

_nanomaterials, 2022, doi:10.3390/nano12142379_

Round 1

Reviewer 1 Report

This review article repots heteroatom doped porous carbon-based nanostructures for electrochemical CO2 reduction as the main application. It covers recent literature about synthesis, characterization, and electrochemical studies of heteroatom doped porous carbon-nanostructures. Overall, this article is quite interesting, but some significant issues need to be addressed before being published in the ‘‘nanomaterials’’. This review article can be improved by addressing the following issues:

1. We suggest that the author make a clear table regarding mono, binary, and ternary doping with their products and durability data (at the end of every topic).

2. In mono doping, "S" and "B" details are very less. Thus, the author must focus on more data and include more data. 

 3. What if the researchers chose S-F, F-P, and S-P combinations in binary doping? (Because this review mainly focused on heteroatoms doping and its effects).

 4. Most of the products in this review are CO (carbon monoxide), which is more toxic than CO2. We recommend that the author include data other than CO products. It will be helpfull for the future researchers to selet materials. 

5. Authors should rewrite the uniform chemical formula in a template-based method using a proper writing manner. 

6. Reference number (78) on new template synthesis need to be addressed few references and there are some inappropriate sentences. Thus, check again and rewrite. (10.1002/advs.202105344; 10.1016/j.apcatb.2021.120405)

7. Authors should prefer a few references that are esy to understand for the template-based method. 

8. The authors should follow the uniform reference format. Please check again all references.

Author Response

Dated: June 23-2022

Reviewer(s)' Comments to Author:

We thank Reviewer #1 for his/her critical and insightful comments on the paper, which significantly helped to improve the quality and clarity of this manuscript. We hope that our revisions and adaptations are adequate and reflect all the suggestions of Reviewer #1. Our detailed responses to reviewer #1 are given below.

Reviewer: 1

Comments:
This review article reports heteroatom doped porous carbon-based nanostructures for electrochemical CO2 reduction as the main application. It covers recent literature about synthesis, characterization, and electrochemical studies of heteroatom doped porous carbon-nanostructures. Overall, this article is quite interesting, but some significant issues need to be addressed before being published in the ‘‘nanomaterials’’. This review article can be improved by addressing the following issues:

Comment 1

We suggest that the author make a clear table regarding mono, binary, and ternary doping with their products and durability data (at the end of every topic).

Reply 1

Thanks for your careful reading and helpful comment. We have added a clear table for detailed comapriosn of the CO2RR perofmace, durability, and products of  mono and binary heteroatom doped HAs-PCs toward CO2RR at the edn of each section, as you can kindly see in the revised manuscript Table 3 and Tbale 4.

Comment 2

In mono doping, "S" and "B" details are very less. Thus, the author must focus on more data and include more data. 

Reply 2

Thanks for your careful reading and helpful comment. We have added more data and details for the mono doing S and B. Kindly see the highlighted sections in ‘’ Metal-free Other Heteroatom (S, F, or B)-doped Porous Carbon Materials.’’

Comment 3

What if the researchers chose S-F, F-P, and S-P combinations in binary doping? (Because this review mainly focused on heteroatoms doping and its effects).

Reply 3

Thanks for your careful reading and helpful comment. Combining S-F, F-P, and S-P in binary doping is essential for enhancing the CO2RR activity and durability. So, in this manuscript, we have summarized and discussed all binary heteroatoms doped porous carbon for CO2RR in the section entitled ‘’2.2 Binary Heteroatom-doped Porous Carbon Materials‘’. However, kindly allow us to emphasize that the fabrication of heteroatoms (mono, binary, and ternary) doped porous carbon nanostructures (HAs-PCs) for CO2RR is rarely reported; only 53 articles have been published in the last decade, according to the web of science. So in this review, we have included all the previous work related to the synthesis of HAs-PCs for CO2RR.

 Comment 4

 Most of the products in this review are CO (carbon monoxide), which is more toxic than CO2. We recommend that the author include data other than CO products. It will be helpful for the future researchers to select materials. 

Reply 4

Thanks for your careful reading and helpful comment. We agree that CO is more toxic than CO2, and conversion of CO2 to other hydrocarbon products like alcohol, formic acid, and C2-C3 compounds is preferred relative to CO. However, previously reported heteroatoms doped porous carbon-based nanostructures mainly reduce CO2 to CO gas, and few articles reported the production of alcohols (i.e., CH3CH2OH and CH3OH), aldehyde,  CH4, format, and formic and we have included them in Table 3. We have included the entire reports related to the production of other hydrocarbons rather than CO in Table 3 in the revised manuscript.

Comment 5

Authors should rewrite the uniform chemical formula in a template-based method using a proper writing manner. 

Reply 5

Thanks for your careful reading and helpful comment. We have fixed the chemical formula in a template-based method as well as fixed all other typos and grammar mistakes, as you can kindly see in the highlighted sections in the revised manuscript

Comment 6

 Reference number (78) on new template synthesis need to be addressed few references and there are some inappropriate sentences. Thus, check again and rewrite. (10.1002/advs.202105344; 10.1016/j.apcatb.2021.120405)

Reply 6

Thanks for your careful reading and helpful comment. We have checked and rewritten the sentences related to Ref.78, as you can see in Lines 244-249. We have also cited the important references you suggested‘’10.1002/advs.202105344 & 10.1016/j.apcatb.2021.120405’’as you can kindly see in Ref.10 and Ref.18.

Comment 7

Authors should prefer a few references that are easy to understand for the template-based method. 

Reply 7

Thanks for your careful reading and helpful comment. We have cited more references related to the template-based method. Kindly see Refs. 58-61

Comment 8

The authors should follow the uniform reference format. Please check again all references

Reply 8

Thanks for your careful reading. We unified the reference format and corrected them according to Nanomaterials style.

Reviewer 2 Report

In this manuscript, the authors systematically review the synthesis methods of heteroatom-doped materials, configurations and effects of heteroatoms, and the enhancement of co-/ternary heteroatom-doped material on electrochemical CO2 reduction. Various template-based methods are introduced for the synthesis of material, including hard-/soft- templates, Ca-based templates, ionic liquids, MOF templates and biomass. The doping atoms are basically classified as N, B, S, P and O and their different combinations are also evaluated. The angle of the review is novel, appealing and inspirable for not only the synthesis of materials but also their reaction mechanism towards CO2 reduction. This is a valuable contribution to the catalytic field, and I would recommend it for publication in Nanomaterials in its present form. 

Author Response

Dated: June 23-2022

Reviewer(s)' Comments to Author:

Reviewer: 2

Comments:
In this manuscript, the authors systematically review the synthesis methods of heteroatom-doped materials, configurations and effects of heteroatoms, and the enhancement of co-/ternary heteroatom-doped material on electrochemical CO2 reduction. Various template-based methods are introduced for the synthesis of material, including hard-/soft- templates, Ca-based templates, ionic liquids, MOF templates and biomass. The doping atoms are basically classified as N, B, S, P and O and their different combinations are also evaluated. The angle of the review is novel, appealing and inspirable for not only the synthesis of materials but also their reaction mechanism towards CO2 reduction. This is a valuable contribution to the catalytic field, and I would recommend it for publication in Nanomaterials in its present form. 

We want to thank Reviewer #2, and we are grateful to Reviewer #2 for accepting our article

mar mistakes in the revised manuscript

Reviewer 3 Report

The manuscript „ Heteroatom Doped Porous Carbon-based Nanostructures for Electrochemical CO2 Reduction “ is quite well written, informative and it contains lots of relevant and comprehensive literature background. Nevertheless, there are minor points that should be addressed by the authors before publishing:

1)      Line 228: a misprint “MnO”

2)      Line 273: “ores”

Author Response

Dated: June 23-2022

We want to thank Reviewer #3 for his/her critical and insightful comments on the paper, which we think significantly helped to improve the quality and clarity of this manuscript. We hope that our revisions and adaptations are adequate and reflect all the suggestions of Reviewer #3. Our detailed responses to reviewer #3 are given below.

Reviewer: 3

Comments:
The manuscript „ Heteroatom Doped Porous Carbon-based Nanostructures for Electrochemical CO2 Reduction “ is quite well written, informative and it contains lots of relevant and comprehensive literature background. Nevertheless, there are minor points that should be addressed by the authors before publishing:

Comment 1

 Line 228: a misprint “MnO”

Reply 1

Thanks for your careful reading and helpful comment. We have fixed it and changed ‘’MnO’’ to ‘’MgO’’. We have also fixed all other typos and grammar mistakes in the revised manuscript

Comment 2

 Line 273: “ores”

Reply 2

Thanks for your careful reading and helpful comment. We have fixed it and changed ‘’ores’’ to ‘’pores’’. We have also fixed all other typos and grammar mistakes in the revised manuscript

Round 2

Reviewer 1 Report

The authors responded appropriately and the present form of the manuscript can be published in Nanomaterials. However, minor mistakes were found in the format of Reference of the manuscript. Please check and correct each one carefully.